# Degradation of Carbendazim by Molecular Hydrogen on Leaf Models

**DOI:** 10.3390/plants11050621

**Published:** 2022-02-25

**Authors:** Tong Zhang, Yueqiao Wang, Zhushan Zhao, Sheng Xu, Wenbiao Shen

**Affiliations:** 1Laboratory Center of Life Sciences, College of Life Sciences, Nanjing Agricultural University, Nanjing 210095, China; 2020116098@stu.njau.edu.cn (T.Z.); 2019116098@njau.edu.cn (Y.W.); 2021816131@stu.njau.edu.cn (Z.Z.); 2Institute of Botany, Jiangsu Province and Chinese Academy of Sciences, Nanjing 210014, China; xusheng@cnbg.net

**Keywords:** hydrogen gas, carbendazim degradation, glutathione metabolism, detoxification system, redox balance

## Abstract

Although molecular hydrogen can alleviate herbicide paraquat and *Fusarium* mycotoxins toxicity in plants and animals, whether or how molecular hydrogen influences pesticide residues in plants is not clear. Here, pot experiments in greenhouse revealed that degradation of carbendazim (a benzimidazole pesticide) in leaves could be positively stimulated by molecular hydrogen, either exogenously applied or with genetic manipulation. Pharmacological and genetic increased hydrogen gas could increase glutathione metabolism and thereafter carbendazim degradation, both of which were abolished by the removal of endogenous glutathione with its synthetic inhibitor, in both tomato and in transgenic *Arabidopsis* when overexpressing the hydrogenase 1 gene from *Chlamydomonas reinhardtii*. Importantly, the antifungal effect of carbendazim in tomato plants was not obviously altered regardless of molecular hydrogen addition. The contribution of glutathione-related detoxification mechanism achieved by molecular hydrogen was confirmed. Our results might not only illustrate a previously undescribed function of molecular hydrogen in plants, but also provide an environmental-friendly approach for the effective elimination or reduction of pesticides residues in crops when grown in pesticides-overused environmental conditions.

## 1. Introduction

Pesticides are indispensable for sustained food production [1]. However, excessive use of pesticides, including fungicides, could result in pesticide pollution of vegetables and environmental contamination. More importantly, fungicides can enter the human body and the food chain through foods, polluted air, and water. Therefore, human health is currently being threatened by the continued use of fungicides [2]. Carbendazim (CAR; methyl 1H-benzimidazol-2-ylcarbamate), a benzimidazole fungicide, is generally used in treatment and control of fungal diseases in vegetables, fruits, flowers, etc. [3]. Since *β*-tubulin is the target of CAR, its antifungal mechanism could be contributed to interfere with the formation of the spindle during the mitosis of pathogenic bacteria [4].

Normally, CAR remains on the plant surface, or is absorbed by plants and then accumulates at the end of the food chain, ultimately posing a serious threat to human health. Importantly, excess CAR has been reported to disrupt the human endocrine system. At low concentrations, this chemical can even damage the mammalian liver, reproductive tissues, and endocrine [5]. Therefore, there is an urgent need to develop effective strategies to reduce CAR residues in agricultural products.

The degradation of CAR is mainly photochemical catalytic degradation and biodegradation, but the process is relatively slow [6]. Thus, research on accelerating the degradation of carbendazim in plants and the environment has attracted more attention, especially with an environmental-friendly approach. For example, previous research discovered the nitric oxide (NO), a natural by-product of nitrogen metabolism [7], can participate in the degradation process of chlorothalonil, a broad-spectrum chlorine fungicide, achieved by brassinosteroid [8].

Hydrogen gas (H_2_) is extremely small and was previously regarded as a relatively inert molecule. During the last decade, the biology of molecular hydrogen in animals has experiencing a surge of discoveries after Japanese scientists discovered that H_2_ is a novel selective antioxidant in rats [9]. In contrast to the clear picture of H_2_ in animals [10], the progress of H_2_ functions in plants is just beginning [11]. Initially in plant biology, H_2_ production is thought to be increased under various abiotic stresses and the normal growth conditions [12,13]. Afterwards, this gas was confirmed to be produced as an obligate by-product of the nitrogenase reaction [14], although other synthetic pathways of H_2_ are still unknown.

Further experiments confirmed that exogenously applied with hydrogen-rich water (HRW) or the fumigation with H_2_ could play vital roles in plant growth, development, and environmental responsiveness [15,16]. The degradation of fungicides remaining on the surface of plants could be accelerated by HRW through brassinosteroids signaling pathway, such as chlorothalonil (CHT) [17]. However, CHT was hardly absorbed by plants compared with CAR [18]. Genetic evidence showed that the expression of the *hydrogenase1* gene (*CrHYD1*) from *Chlamydomonas reinhardtii* not only increased endogenous H_2_ production, but also confers Arabidopsis tolerance against salinity [19] and drought stress [20]. Additionally, H_2_ could influence nitric oxide [21], abscisic acid [22], auxin [23], melatonin [19], and glutathione [24] signaling in plants.

This study wants to elucidate whether the application of H_2_ could stimulate the degradation of CAR, a systemic fungicide applied in agriculture. First, genetic and physiological evidence showed that endogenous H_2_ plays an important role in regulating CAR degradation in plant leaves through intensifying glutathione synthesis. Based on these results, we further suggest that CAR degradation might be controlled by genetic manipulation of endogenous H_2_ or exogenously applied with H_2_. Given the inherent toxicity of carbendazim and the intentional release into the environment [25], our results may contribute to the application of molecular hydrogen in agriculture and human health.

## 2. Results and Discussion

### 2.1. Exogenous H_2_ Control of CAR Degradation

Though molecular hydrogen could enhance the metabolism of CHT in plants [17], the adsorption characteristics between CHT and CAR were entirely different. CHT was just attached to the leaf surface and hardly absorbed by plants, while CAR was absorbed into the plants and then exerted the medicinal effect. Thus, whether or how molecular hydrogen affects the metabolism of systemic fungicides in plants is still not clarified. To explore the problem above, tomato plants with six leaves were selected for the following experiments. Firstly, time curve analysis showed that after CAR addition, the absorption of this fungicide in tomato leaves was rapidly increased during the first 24 h period, followed by the progressive decline until 96 h (Appendix A). Therefore, tomato seedlings after treated with CAR for 24 h were used to assess the effect of exogenous molecular hydrogen on CAR degradation.

After the above CAR-treated seedlings were shifted to no CAR condition (CAR→dH_2_O), its residue in leaves was moderately decreased during a 96-h period (Figure 1A). This result was consistent with the former findings [26], reflecting the photochemical catalytic degradation and biodegradation of CAR. Meanwhile, we observed that CAR addition for 24 h could rapidly stimulate the H_2_ production, and reach the peak after 12 h (Figure 1B). These results demonstrated that the H_2_ concentration was increased in response to CAR.

Subsequent experiments were performed to investigate whether this H_2_ increase was directed towards counteracting CAR phytotoxicity. When treated with saturated HRW, which contained 0.78 mM H_2_ produced by the electrolysis method, the degradation of CAR in tomato seedling leaves was remarkably intensified approximately in a time-dependent fashion (Figure 1A). For example, molecular hydrogen treated tomato leaves displayed residues of 86.2%, 68.1%, 59.6%, 52.2%, and 46.6% compared to those of the non-HRW treatment at 12, 24, 48, 72, and 96 h, respectively. Meanwhile, HRW addition could rapidly increase endogenous H_2_ content in CAR-treated tomato leaves, reaching the peak after 3 h (Figure 1B). Similar results were discovered in seedlings of rice [27], alfalfa [28], and Arabidopsis [22] when treated in stressed conditions. Therefore, we further confirmed that endogenous molecular hydrogen metabolism might be regulated by environmental stimuli [29].

To assess the above results, six common crops that usually use CAR [30], including radish, cucumber, rice, rapeseed, alfalfa, and pepper, were selected for the further experiments. As shown in Appendix A, it was clearly illustrated that saturated HRW could remarkably promote the degradation of CAR in the above-mentioned crops, implying that molecular hydrogen control of CAR degradation might be a universal event.

### 2.2. Exogenous H_2_ Did Not Alter the Antifungal Effect of CAR

Whether molecular hydrogen could influence the bactericidal effect of CAR is another concern. To answer this scientific question, *Alternaria solani*, which can cause several diseases on foliage, basal stems of seedlings, and fruits of tomato [31], were applied in PDA culture medium in the presence or absence of either HRW or CAR. After 10 days of inoculation, it was clearly observed that the colony diameter was differentially inhibited by CAR, when its concentrations were ranging from 2.62 to 20.92 mM (Figure 2A,B). Importantly, the above inhibition achieved by CAR was not altered regardless of HRW addition. Thus, HRW might not repress the growth and development of pathogenic bacteria. Consistently, no significant differences in the disease rate of leaf were observed in fungus-infected tomato seedlings at the six-leaves stage with or without HRW treatments (Figure 2C–F). The above results clearly showed that the antifungal effect of CAR was not altered by HRW. It was an interesting finding. This result further confirmed the possibility that the detoxification pathway triggered by molecular hydrogen might be mediated by a specific route, rather than binding tubulin molecule and inhibiting its role in microtubule assembly. Certainly, these parallel events should be carefully explored.

### 2.3. Glutathione Involvement in Molecular Hydrogen Control of CAR Degradation

Since glutathione (GSH) is identified as an important detoxification compound and defensive signaling molecule in both human disease responses and plant adaptation to environmental stresses [32], the changes in GSH-metabolism and detoxification genes expression were analyzed by using RT-qPCR. As shown in Figure 3, CAR stimulated the expression of *GS* (encoding glutathione synthetase), *GST* genes (encoding glutathione *S*-transferase), *GPX* (encoding glutathione synthetase), *ABC2* (encoding ABC transporters), and *CYP724B2* (encoding one of the members of brassinosteroids) in the plants [33,34]. It could be speculated that the enhancement of the above gene expression following exposure to CAR might be an initial defense response for intensifying the GSH-metabolism and detoxification systems. Afterwards, we observed that HRW addition could further stimulate the expression of the above genes compared to the dH_2_O control samples.

Subsequently, endogenous GSH levels were monitored under the identical conditions. Although the pretreatment with CAR for 24 h could not obviously influence the changes of reduced GSH levels [34], the addition of HRW did trigger GSH production, peaking at 24 h after treatment, then followed by a gradual decrease, but still above the basal level (Figure 4A). Unlike the responses in reduced GSH, oxidized GSH (GSSG) production in tomato seedling leaves was stimulated by CAR, which was further maintained by HRW up to 48 h after treatment compared to non-HRW control samples (Figure 4B). A similar change in GSH pool, especially GSSG, achieved by CAR, was previously reported in tomato plants [26]. The above results clearly showed that an increase in the concentration of GSH might be one of the earliest responses participating in the signaling transduction triggered by molecular hydrogen in tomato leaves. Subsequent results revealed that GSH addition could improve photosynthetic characteristics. As expected, treatment with CAR could severely inhibit the values of the net photosynthetic rate (Pn), the maximum PSII quantum yield (Fv/Fm), and photochemical quenching coefficient (qP), reflecting the phytotoxicity of CAR (Appendix AA–C), which was also accompanied with the increased lipid peroxidation in tomato seedling leaves (TBARS; Appendix AD). After GSH addition, the above parameters could be differentially improved, especially with 5 mM GSH exhibiting the maximal responses, which was used subsequently.

To assess the contribution of endogenous GSH in the response of CAR degradation achieved by molecular hydrogen, _L_-buthionine-sulfoximine (BSO), an effective inhibitor of *γ*-glutamylcysteine synthetase (*γ*-ECS) responsible for GSH synthesis [35,36], was applied individually or simultaneously. Further experiments showed an obvious increase of endogenous GSH content, and importantly, a significant reduction in CAR content, after the addition of GSH (Figure 4C,D), mimicking the responses of HRW. By contrast, the above-mentioned results were remarkably abolished by BSO, reflecting the important function of endogenous GSH. Alone, BSO could result in the accumulation of CAR and in the depletion of GSH.

The GSH biosynthesis induction by molecular hydrogen was previously observed in the stressed conditions [37] or during lateral root development [24]. To further decipher the protective role of GSH in molecular hydrogen control of CAR degradation, GSH metabolism was examined. As shown in Figure 4E, in the presence of CAR, both HRW and especially GSH could significantly enhance the ratio of GSH/GSSG. Unlike the effect of exogenous GSH, the above response of HRW might be caused by the increased activity of *γ*-ECS (Figure 4F), an important enzyme catalyzed GSH synthesis [36]. These indicated the stimulation of GSH production achieved by molecular hydrogen via intensifying *γ*-ECS activity, and similar results were found in both animals and plants, especially under environmental stimuli [37]. Consistently, the above responses in GSH/GSSG and *γ*-ECS activity conferred by HRW were abolished by the removal of endogenous GSH in the presence of BSO, reflecting that the HRW control of the response is *γ*-ECS-dependent. When applied alone, BSO did result in negative effects.

Both glutathione *S*-transferase (GST) and glutathione reductase (GR) are two enzymes related to GSH metabolism. Between these, GR is responsible for reducing GSSG back to GSH [38]. Particularly, ample evidence showed that GSH control of organic compounds detoxification is closely associated with the reaction catalyzed by GST, and its product, GSH conjugates, could be transported into the vacuole [39]. Accordingly, the changes in GR and GST were investigated and compared in the presence or absence of HRW or GSH, with or without the addition of BSO.

For GST, further results clearly revealed that both HRW and GSH intensified its activities, which were further impaired by BSO (Figure 4G). Importantly, these data were correlated to changes in CAR residues (Figure 4C). These might be explained by the fact that CAR residues might be bound to GSH catalyzed by enhanced GST activity to form less reactive and toxic conjugates for subsequent transportation and degradation [40]. Similar changes were observed in the contents of nonprotein thiols (NPT; Appendix AA). Since NPT were closely associated with GSH conjugates, we further deduced that the stimulation of NPT synthesis achieved by molecular hydrogen might reflect the decomposition products of pesticide to some extent [41]. After the addition of CAR, unlike the response of GSH, HRW could remarkably increase the activity of GR (Figure 4H). These changes also matched with the GSH content and the ratio of GSH and GSSG, further supporting the important function of endogenous GSH in the molecular hydrogen control of CAR degradation.

### 2.4. Transcriptional Regulation of GSH-Metabolism and Detoxification Genes

To probe the molecular mechanism underlying molecular hydrogen control of GSH synthesis, RT-qPCR was applied. As shown in Figure 5A–C, HRW strongly stimulated the expression of *GR*, *GSH1* (encoding *γ*-ECS) [42], and *GST2* in the presence of CAR, all of which were matched with the increased activities of GR (Figure 4H), *γ*-ECS (Figure 4F), and GST (Figure 4G), thus promoting endogenous GSH production in CAR-treated tomato seedling leaves (Figure 4A).

Meanwhile, three detoxification genes, including *CYP72A*, *ABC3*, and *ABC4*, were selected to evaluate their involvement of exogenous H_2_ control of CAR degradation. Among these, *CYP72A* encodes one of the members of cytochrome P450s, which are indispensable for hormone synthesis and detoxification in both animals and plants [43]. ATP-binding cassette (ABC) transporters also participated in the detoxification of environmental pollutants [44,45]. Similarly, we further observed that in the presence of CAR, the expression of *CYP72A* (Figure 5D), *ABC3* (Figure 5E), and *ABC4* (Figure 5F) were significantly intensified by HRW, which were also correlated with the increased degradation of CAR (Figure 1A).

### 2.5. CAR-Triggered Redox Imbalance Was Abolished by HRW

For pesticide-induced phytotoxicity, ample evidence revealed that insecticides could influence redox balance, including resulting in reactive oxygen species (ROS) production and oxidative damage, in both animal and plant cells [46]. RT-qPCR results showed that *CAT1* (encoding catalase), *G-POD* (encoding peroxidase), *APX1* (encoding ascorbate peroxidase), *Cu/Zn-SOD* (encoding superoxide dismutase), *MDHAR* (encoding monodehydroascorbate reductase), and *DHAR* (encoding dehydroascorbate reductase) genes (Zhou et al., 2017) were up-regulated after applying CAR (Appendix A). Furthermore, the combined treatment of CAR and HRW induced the transcription of the above-mentioned genes more strongly than CAR or HRW alone, indicating an additive effect of CAR and HRW on the induction.

Subsequent results showed that in the presence of either HRW or GSH, both H_2_O_2_ and O_2_^−^ distribution (two important components of ROS; Appendix AB,C) and TBARS accumulation (an index related to oxidative damage; Appendix AD) were found to be abolished in CAR-treated tomato leaves, indicating the reconstructing redox homeostasis achieved by both HRW and GSH. These effects were sensitive to the removal of endogenous GSH by the addition of BSO, thus further emphasizing the participation of GSH. Similar tendencies were found in the changes of four antioxidant enzymes activities, including superoxide dismutase (SOD; Appendix AE), catalase (CAT; Appendix AF), ascorbate peroxidase (APX; Appendix AG), and guaiacol peroxidase (POD; Appendix AH).

### 2.6. Genetic Evidence Revealed That Endogenous Molecular Hydrogen Can Positively Influence Carbendazim Degradation via GSH

Exogenously applied molecular hydrogen may not faithfully mimic the function of endogenous H_2_ [22]. Thus, the genetic-based approach, using genetic materials with altered endogenous H_2_ levels, is likely to be more accurate for dissecting the nature function of molecular hydrogen, although its metabolism is still not fully elucidated in plants [11].

In the previous reports, the expression of *hydrogenase1* gene from *Chlamydomonas reinhardtii* (*CrHYD1*) in Arabidopsis represents an interesting method to assess the functions of endogenous production of H_2_ in plant cells, since molecular hydrogen derived from *CrHYD1* expression could modify the stomatal closure [20] and improve salinity tolerance [19]. Here, compared to those in the wild-type (WT) with or without being transformed with the empty vector (EV), six *CrHYD1* lines not only showed increased H_2_ production in the presence of CAR (Appendix A), but importantly, displayed the lower residues of CAR in leaves, especially *CrHYD1-3*, *CrHYD1-5*, and *CrHYD1-6* (Figure 6A), and similar beneficial roles of endogenous H_2_ were discovered according to the changes in stomatal bioassays [20] and salinity phenotypic analysis [19]. Under the identical treatments, contrasting changes in endogenous GSH were also observed (Figure 6B), thus reflecting the possible negative correlation between CAR residues and GSH contents. This deduction was further supported by the ones exogenously applied with either GSH or BSO after the spraying with CAR in the above transgenic lines and wild-type plants. For example, after the addition of GSH, the decreased CAR residues and increased GSH contents observed in transgenic lines were differentially intensified. Contrasting results were found when BSO was applied, which could inhibit endogenous GSH biosynthesis. The above results clearly provided genetic evidence, showing that endogenous H_2_ is an endogenous regulator for CAR degradation in a GSH-dependent fashion.

In summary, our results clearly demonstrated that either exogenously applied or endogenously increased molecular hydrogen can decrease CAR residues in plant leaves. The present findings also indicated the possible role of GSH-dependent pathway in the detoxification of CAR, although a corresponding mechanism has not been fully elucidated.

Although several reports discovered that both brassinosteroids [40] and melatonin [26] could help plants degrade pesticides, the application with HRW without using the above conventional chemical additives might be more environmentally friendly for sustainable agriculture. More importantly, this study opened a new way for molecular hydrogen application to degrade the systemic fungicide CAR in a crop.

## 3. Materials and Methods

### 3.1. Chemicals

_DL_-Buthionine-*S* (BSO; CAS 83730-53-4; purity ≥ 97%) and reduced glutathione (GSH; CAS 27025-41-8; purity ≥ 98%) were purchased from Sigma-Aldrich (St. Louis, MO, USA). Carbendazim (CAR; 50% active ingredient) was obtained from Yingkouleike (Liaoning, China).

### 3.2. Plant Material, Growth Conditions, and Treatments

According to the previous methods [47], five-day-old tomato (*Solanum lycopersicum* L. cv. baiguoqiangfeng) seedlings were transferred to a flowerpot containing a mixture of peat and vermiculite (3:1; *v*/*v*) and were cultivated in a growth chamber with a light intensity of 200 μmol m^−2^ s^−1^ under a light cycle of 14 h of light and 10 h of dark at 24 °C. Full-strength Hoagland’s nutrient solution was replaced every two days until six true leaves were grown.

The *CrHYD1* transgenic strain expressed the *hydrogenase1* gene from *C. reinhardtii* in Arabidopsis plants under the control of the cauliflower mosaic virus (*CaMV*) 35S promoter [20]. Seven-day-old Arabidopsis seedlings including wild-type, control plants transformed with the empty vector (EV), and six *CrHYD1* transgenic lines (*CrHYD1-1*, *CrHYD1-2*, *CrHYD1-3*, *CrHYD1-4*, *CrHYD1-5*, and *CrHYD1-6*) were cultivated in a growth chamber with a light intensity of 150 μmol m^−2^ s^−1^ under a light cycle of 16 h of light and 8 h of dark at 23 °C for 6 weeks.

In order to elucidate the interaction between molecular hydrogen and CAR degradation (Figure 1) and whether GSH metabolism is stimulated by molecular hydrogen (Figure 4), CAR was dissolved in distilled water to prepare a solution of 10.46 mM (the concentration used in practical applications) and further sprayed on tomato leaves [34], with 20 mL CAR used for each potted plant. After 24 h treatment, the above potted plants were divided into two groups and placed in two identical trays (20–40 pots for each). The two trays were filled with 500 mL dH_2_O or HRW, respectively.

In order to assess whether the CAR degradation achieved by molecular hydrogen was related to GSH metabolism (Figure 4 and Figure 6), or altered the redox balance (Appendix A), 10.46 mM CAR was used, with 20 mL CAR sprayed for each potted plant. After 24 h of treatment, the above potted plants were divided into six groups, and respectively placed in six identical trays (20–40 pots for each).

Four groups were respectively filled with 500 mL distilled water, and each potted plant was sprayed with 5 mL distilled water containing an equal ratio of organic solvent (CAR→dH_2_O/Con), or sprayed with 5 mL 5 mM GSH (CAR→GSH), or sprayed with 5 mL 5 mM GSH and 5 mL 1 mM BSO (CAR→GSH + BSO), or sprayed with 5 mL 1 mM BSO (CAR→BSO).

Two other groups were respectively filled with 500 mL HRW, and each potted plant was sprayed with 5 mL distilled water containing an equal ratio of organic solvent (CAR→HRW), or sprayed with 5 mL 1 mM BSO (CAR→HRW + BSO).

### 3.3. Preparation of Hydrogen-Rich Water

According to the previous method [22], the saturated hydrogen-rich water (HRW) was prepared by using a SHC-300 H_2_ generator (Saikesaisi Hydrogen Energy Co., Ltd., Jinan, China). The H_2_ concentration in the above HRW was about 0.78 mM, and this saturated solution was applied subsequently.

### 3.4. Determination of H_2_ Content

Endogenous H_2_ content was measured with headspace gas chromatography (Tianmei GC7900 equipped with a thermal conductivity detector, Tianmei Scientific Instrument Co., Ltd., Shanghai, China) as described previously [48].

### 3.5. Determination of GSH Content by UPLC Analysis

According to previous methods [49], the GSH contents in leaves were determined with ultra-high performance liquid chromatography (UPLC; Agilent 1290, Agilent, Palo Alto, CA, USA). Total GSH and GSSG contents were determined by UPLC, and reduced GSH content was estimated from the difference between total GSH and GSSG.

### 3.6. Determination of CAR Residues in Tomato Leaves

CAR residues were determined by UPLC (Agilent 1290, Agilent, Palo Alto, CA, USA) [26]. The wavelength of detection was 280 nm, and the injection volume was 2 μL.

### 3.7. Indoor Toxicity Assessment and Experiment of Early Blight Resistance

According to a previous method [50], indoor toxicity has been determined. *Alternaria solani* (Shanghai Bioresource Collection Center, SHBCC, Shanghai, China) was inoculated into the center of potato dextrose agar (PDA) mediums containing 0, 2.62, 5.23, 10.46, or 20.92 mM CAR at 28 °C. Organic membrane filters with the pore size of 0.22 μm were used to filter HRW and distilled water, which were added to the culture medium. The colony diameter was determined after the incubation for 10 days.

After *Alternaria solani* was propagated on PDA mediums for 10 days, 2 mL of sterile water was added, and the spore was slowly scraped with a microscope slide. After inoculation, the spore concentration of the suspension was adjusted to 1 × 10^4^ conidia mL^−1^, which was sprayed on tomato seedlings at the six-leaves stage [51]. The disease rate of leaves was calculated after 15 d of treatment [52].

### 3.8. Analysis of the GSH Cycle

According to the previous method [53], the activity of GR (glutathione reductase) was analyzed by detecting the rate of decrease in the absorbance of 340 nm. GST (glutathione *S*-transferase) and *γ*-ECS (*γ*-glutamylcysteine synthetase) activities were assayed with GST and *γ*-ECS activity kit (Nanjing JianCheng Bioengineering Institute, Nanjing, China). One unit (U) of *γ*-ECS activity was defined as the production of 1 μmol inorganic phosphorus per milligram of tissue protein per hour.

### 3.9. RT-qPCR Determination of Transcript Levels of Genes

After the isolation of total RNA and cDNA synthesis, real-time quantitative PCR (RT-qPCR) was performed [54], and the gene-specific primers were listed in Appendix A. *GADPH* was used as an internal control gene. Three technical replicates of RT-qPCR were performed per gene-specific primers pair.

### 3.10. Statistical Analysis

Statistical analysis was performed using IBM SPSS Statistics 16.0 software. Statistical significance was analyzed by ANOVA analysis in combination with Tukey’s multiple test (*p* < 0.05) or independent-sample *t*-test. ** and *** indicate significant difference results at *p* < 0.01 or *p* < 0.001.

## 4. Conclusions

The above molecular and genetic evidence demonstrated that H_2_ could promote the CAR degradation in plants. GSH operates a downstream molecule functioning in the above process. In addition, molecular hydrogen could effectively enhance plant antioxidant capacity, thus reestablishing redox balance. Importantly, the results from this study will assist and promote further efforts to bring the findings of basic hydrogen biology research to hydrogen-based agriculture, thus meeting the dietary needs of 7.6 billion people in a healthy and sustainable manner.

## Figures and Tables

**Figure 1 plants-11-00621-f001:**
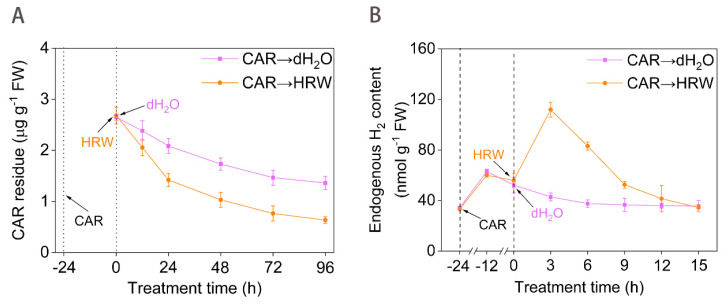
A possible link between the degradation of CAR and molecular hydrogen. Tomato seedlings at the six leaves stage were exposed to 10.46 mM CAR for 24 h, followed by the treatment with dH_2_O and HRW for 24 h. Afterwards, time course of changes in CAR degradation curve (**A**) and endogenous H_2_ contents (**B**) were determined. Error bars represent the standard deviation (SD; n = 3).

**Figure 2 plants-11-00621-f002:**
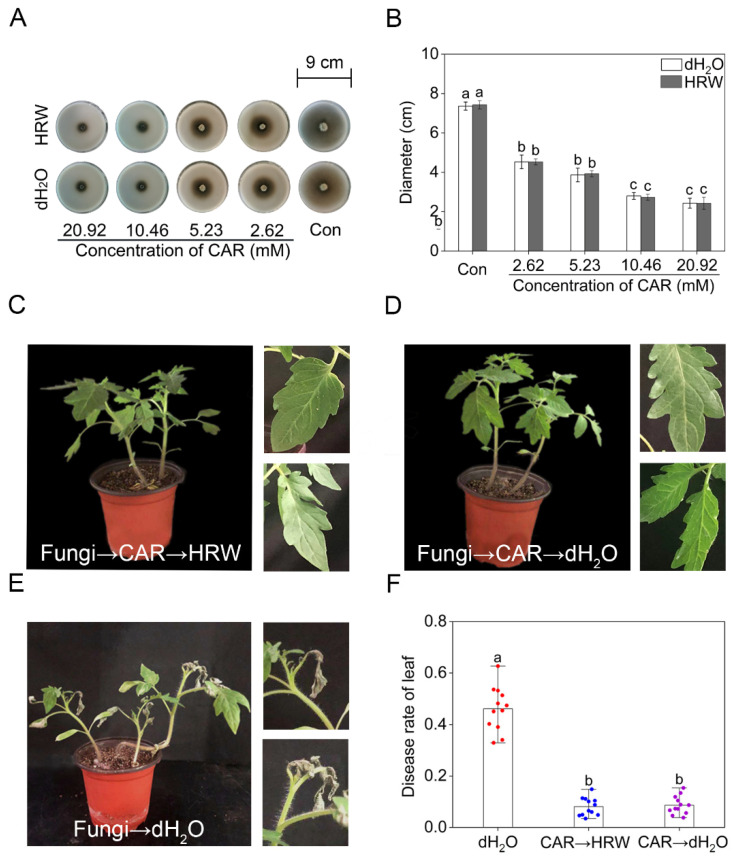
The antifungal effect of CAR was not affected by HRW. *Alternaria solani* was cultured in PDA culture mediums. After treatment for 10 days, the representative phenotypes were photographed (**A**). The diameter of the colony was counted (**B**). Tomato seedlings at the six–leaves stage were divided into three groups. The first and second groups were treated with 10.46 mM CAR for 24 h after inoculated by *Alternaria solani*. Afterwards, the two groups were respectively treated with HRW (CAR→HRW) or distilled water (CAR→dH_2_O) every two days. The third group was inoculated with *Alternaria solani* for 24 h and treated with distilled water (dH_2_O) every two days. After 15 days, the representative phenotypes were photographed (**C**–**E**). The diseased leaves rate of three different treatments was counted (**F**). The error bars represent the SDs (n = 3 for (**B**), n = 12 for (**F**)), and the different letters indicate significantly different values (*p* < 0.05 according to Tukey’s multiple test).

**Figure 3 plants-11-00621-f003:**
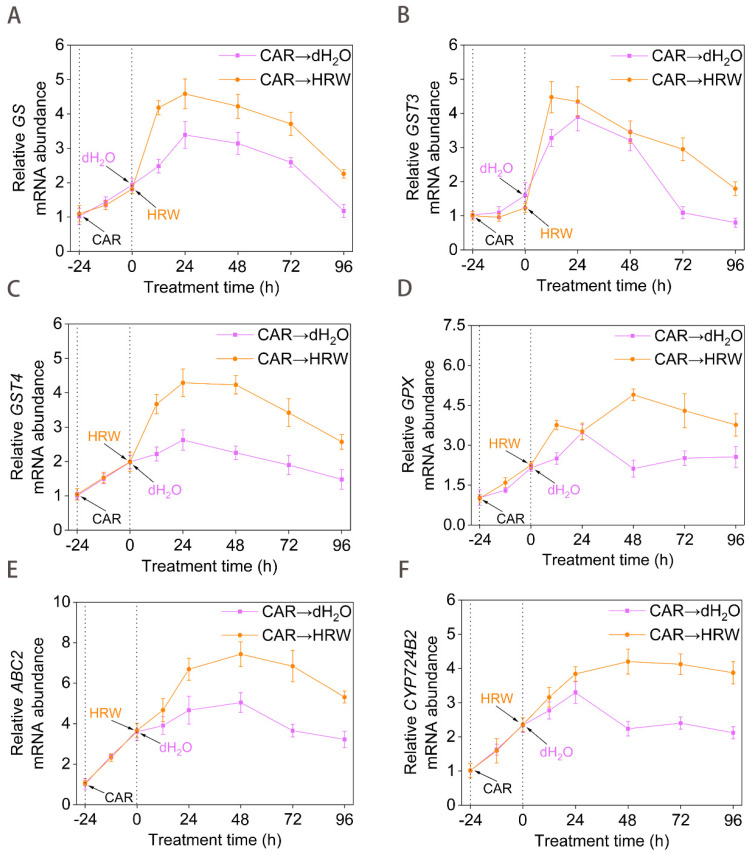
Time curve of GSH−metabolism and detoxification genes expression. Tomato seedlings at the six leaves stage were exposed to 10.46 mM CAR for 24 h, followed by the treatment with dH_2_O or HRW for 24 h. Afterwards, the time course of changes in *GS* (**A**), *GST3* (**B**), *GST4* (**C**), *GPX* (**D**), *ABC2* (**E**), *CYP724B2* (**F**) genes expression was determined. Error bars represent the standard deviation (SD; n = 3).

**Figure 4 plants-11-00621-f004:**
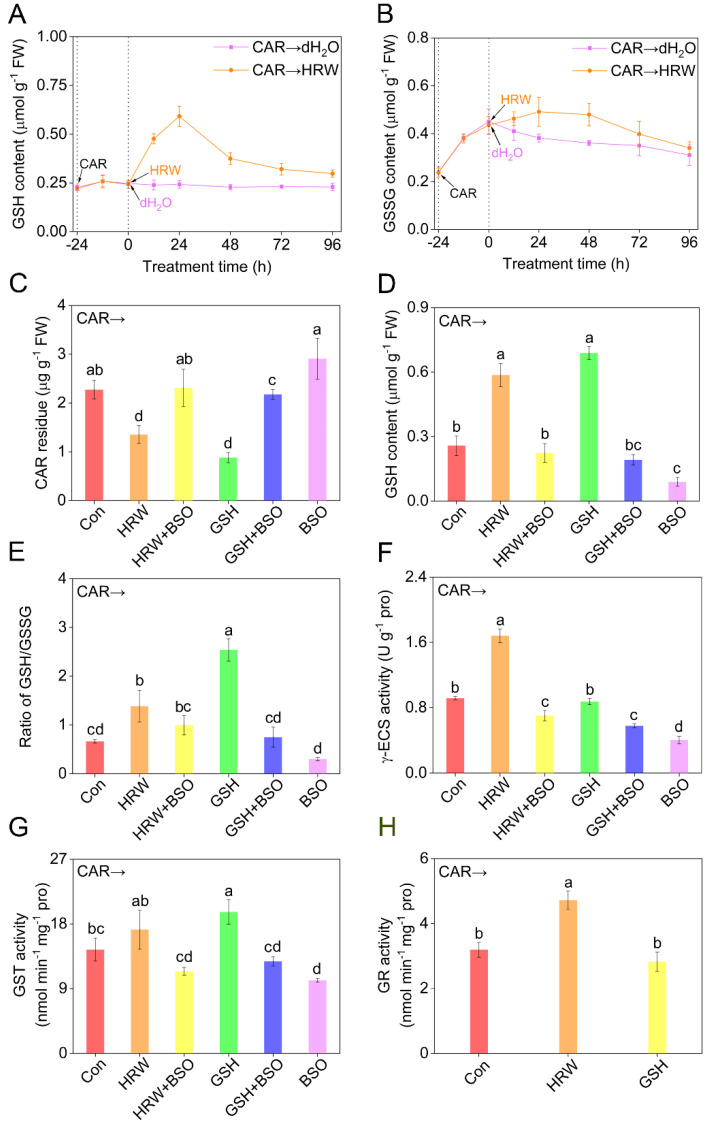
Changes in glutathione−related metabolism and CAR residues. Seedlings at the six−leaves stage were exposed to 10.46 mM CAR for 24 h, followed by the treatment with dH_2_O (Con), HRW, GSH, or BSO, alone or their combinations. Afterwards, time−course changes in endogenous reduced GSH (**A**) and GSSG (**B**) contents were analyzed. After treatments for 24 h, CAR residue (**C**), reduced GSH content (**D**), the ratio of GSH/GSSG (**E**), and *γ*-ECS (**F**), GST (**G**), and GR (**H**) activities in tomato leaves were determined. Error bars represent the standard deviation (SD; n = 3). Bars with different letters are significantly different (*p* < 0.05) according to Tukey’s multiple test.

**Figure 5 plants-11-00621-f005:**
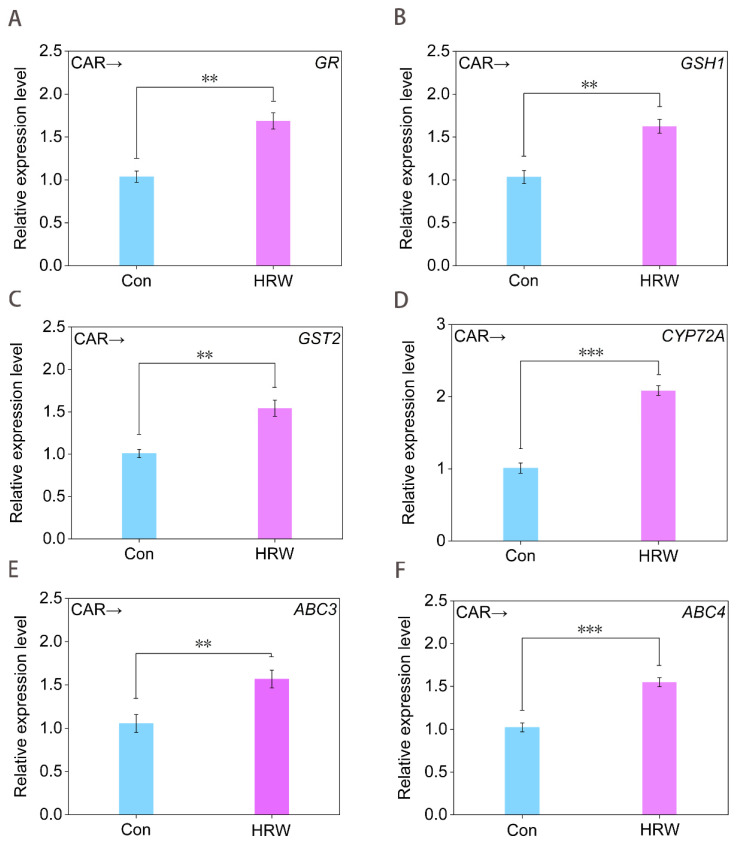
Changes in the transcription levels of GSH−dependent genes and detoxification genes. Tomato seedlings at the six−leaves stage were treated with 10.46 mM CAR for 24 h, followed by treatment with dH_2_O (Con) or HRW. After 24 h of treatment, *GR* (**A**), *GSH1* (**B**), *GST2* (**C**), *CYP72A* (**D**), *ABC3* (**E**), and *ABC4* (**F**) transcript levels were analyzed by RT-qPCR. Error bars represent the standard deviation (SD; n = 3). ** and *** indicate significant difference results at *p* < 0.01 or *p* < 0.001 analyzed by independent-sample *t*-test.

**Figure 6 plants-11-00621-f006:**
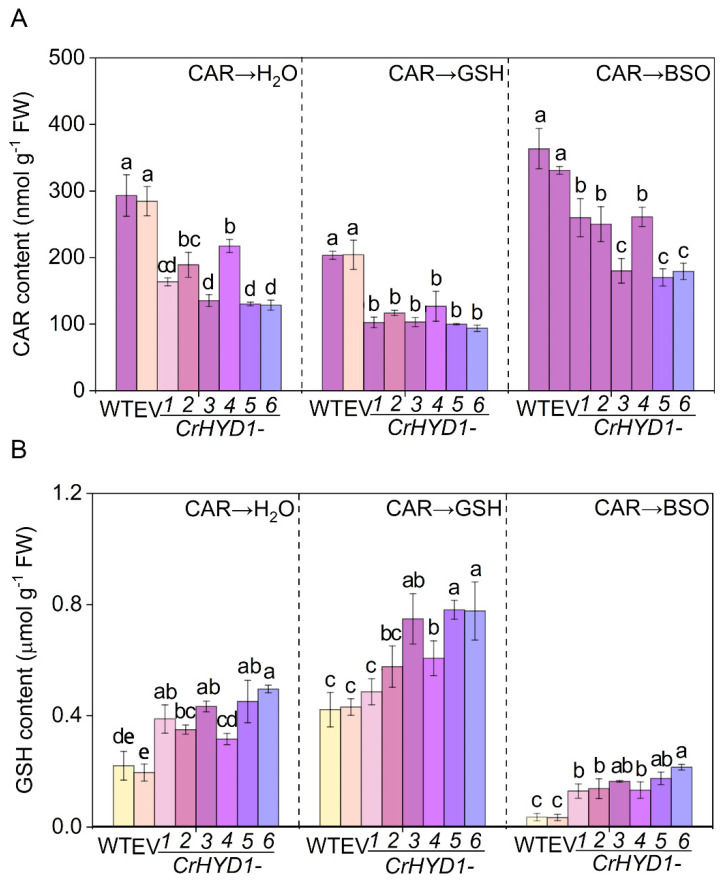
Genetic evidence showing the important role of endogenous molecular hydrogen in promoting the degradation of CAR via GSH synthesis. After *Arabidopsis* seedlings were sprayed with 10.46 mM CAR for 24 h, seedlings were treated with dH_2_O (Con), GSH, or BSO for 24 h. Afterwards, CAR residues (**A**) and GSH content (**B**) were measured. The error bars represent the SDs (n = 3; 20 plants/treatment/repeat). Different letters indicate significantly different values (*p* < 0.05 according to Tukey’s multiple test).

## Data Availability

Data is contained within the article or Appendix A.

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
