# Peer review of "Degradation of Carbendazim by Molecular Hydrogen on Leaf Models"

_plants, 2022, doi:10.3390/plants11050621_

Round 1
Reviewer 1 Report
Review of the paper entitled “Degradation of carbendazim by molecular hydrogen on leave models” by Tong Zhang, Yueqiao Wang, Zhushan Zhao, Sheng Xu, Wenbiao Shen
Carbendazim (CAR) is a systemic, broad-spectrum benzimidazole fungicide. The fungicide is used to control plant diseases mainly in cereals and fruits. The complete degradation of organic compounds (mineralization) in plant cells takes place rarely, while the products of biotransformation, which are fragments of parent molecules, can be used, for example, in the synthesis of amino acids or coupled with natural plant components.
Such an endogenous chemical compound, extremely important in coupling reactions with detoxified xenobiotics, is glutathione (GSH). It is a tripeptide consists of glutamic acid (Glu), cysteine (Cys) and glycine (Gly) residues, in which the first 2 amino acids are bonded through the γ-carboxyl group of Glu. By the way, it is worth adding that some plants have natural GSH analogs instead of Gly the rest of ß-alanine (legumes) or serine (Ser) (cereals). The so-called homoglutathione and hydroxymethylglutathione, respectively may be present in the cell in addition to or in place of GSH. Due to the Cys thiol group, these peptides are a source of non-protein sulfur.
Non-enzymatic nucleophilic coupling of GSH with xenobiotic however, there is a slow reaction, its acceleration occurs with the participation of an enzyme that is glutathione S-transferase (GST, E.C.2.5.1.18). GST is widespread in aerobic organisms, it is found in the microsomal fraction of the cell, cytosol, mitochondria, nucleus and occurs in various isoenzymatic forms.
As in animals, GSH is synthesized in plants from its constituent amino acids by two ATP-dependent steps. The responsible enzymes are found both in chloroplasts and in the cytosol. They are γ-glutamylcysteine synthetase (EC 6.3.2.2; γ-ECS) and glutathione synthetase (EC 6.3.2.3; GS). Their activity depends inter alia, the availability of the constituent amino acids.
Liu et al. demonstrated that the addition of molecular hydrogen (H2) could trigger higher transcript levels of SlGSH1 and SlGSH2, encoding γ-ECS and GS in the tomato cells, which means that H2 enhanced GSH synthesis in these cells (Liu F, Lou W, Wang J, Li Q, Shen W. Glutathione produced by γ-glutamyl cysteine synthetase acts downstream of hydrogen to positively influence lateral root branching. Plant Physiol Biochem. 2021 Oct;167:68-76. doi: 10.1016/j.plaphy.2021.07.034.).
The results obtained by the Authors of the paper I reviewed indicated that exogenously applied or endogenously increased H2 decreased CAR residues in plant leaves. The Authors' results are also consistent with the observations of other authors that molecular hydrogen induces syntheses of GSH in plant cells.
On the basis of the obtained results, the Authors formulated a conclusion that GSH works as a substrate for CAR conjugation.
In conclusion, the Authors suggest that their results “will assist and promote further efforts to bring findings of basic hydrogen biology research to hydrogen-based agriculture, thus meeting the dietary needs of 7.6 billion people in a healthy and sustainable manner”.
The paper is well written and the topic is interesting, however, I have a few comments.
- It seems that the CAR concentrations used by the Authors are high. Maximum pesticide residue limits (MRLs) have reduced since discovering its harmful effects. The MRLs are now between 0.1 and 0.7 mg/kg. Admittedly, the Authors wrote: “the concentration used in practical applications”, nevertheless, I would ask the Authors for a broader comment regarding the use of concentrations of CAR.
- What kind of solvent did the Authors use to make the CAR solution?
- The Authors cited their previous paper, viz: Wang J, Jiang Y, Chen S, Xia X, Shi K, Zhou Y, Yu Y, Yu J. The different responses of glutathione-dependent detoxification pathway to fungicide chlorothalonil and carbendazim in tomato leaves. Chemosphere. 2010, 79(9):958-65. doi: 10.1016/j.chemosphere.2010.02.020. In this paper the Authors also investigated the detoxification possibilities of the CAR in tomato leaves. Based on the results obtained here, the authors formulated the following conclusion: these results suggest that GSH-dependent pathway plays an important role in the chlorothalonil (CHT) detoxification but not in the CAR detoxification in tomato leaves”. In my opinion the Authors should comment in the Discussion on the results obtained in their previous experiments with the results obtained in the paper I have now reviewed, as the conclusions of the two papers are contradictory.
- S-conjugation of xenobiotics with GSH is a detoxification reaction. On the other hand, it is known that several classes of compounds are converted by glutathione conjugate formation to toxic metabolites (Koob M, Dekant W. Bioactivation of xenobiotics by formation of toxic glutathione conjugates. Chem Biol Interact. 1991;77(2):107-36. doi: 10.1016/0009-2797(91)90068-i). In addition, the S-conjugates can be inhibitors of GSH metabolism enzymes (Coleman Jod, Blake-Kalff MMA, Davies TGE. Detoxification of xenobiotics by plants: chemical modification and vacuolar compartmentation. Trends Plant Sci 1997; 2,4: 144.151; Kreuz K, Tommasini R, Martinoia E. Old enzymes for a new job. Herbicide detoxification in plants. Plant Physiol 1996; 111: 349.353). Do the Authors know about the adverse effects of glutathione S-conjugate derived from CAR on the structure and metabolism of a plant cell?
- The glutathione S-conjugates in animal organisms are removed from the cell and excreted out of the body. In plants, instead of excretion, conjugates of xenobiotics are transported to the vacuole. In both cases (in animals and in plants) this leads to a depletion of the intracellular pool of GSH, which is unfavorable. What is the Authors' task on this subject?

Reviewer 2 Report
I think this is a very good paper suitable for publication in this journal. Only one comment: Some statistical analysis was done by Duncan's test. This method is now old fashioned, and said to include some error. As this paper is of good quality, I advise the authors to change to more reliable test, such as Tukey or Scheffe test.

Reviewer 3 Report
The manuscript entitled “Degradation of carbendazim by molecular hydrogen on leave models " is based on original research experiment and the presented results therein broaden the knowledge of plant sciences. To describe genetic and physiological role of endogenous H2 in regulating CAR degradation in plant leaves through intensifying glutathione synthesis was the main aim of this work. The scope of work includes the performance of experiment in controlled conditions, during which determination of GSH content by UPLC analysis, CAR residues in tomato leaves, indoor toxicity assessment and RT-qPCR determination of transcript levels of genes were obtained.
There is no doubt that this work is in the scope of Plants journal. The publication presents interesting and important studies. The work delivers some interesting results and can be the important source of valuable information.
The introduction is properly composed. The materials and methods section contains the basic requested elements and provide information about the experimental preparations, analyses and growth conditions. In this point it should be noted, that authors put in the MM chapter right after the introduction, but in this journal MM should be after the discussion. The data analysis is generally properly provided. The results show valuable information. The obtained data are discussed sufficiently.
However, the authors made some shortcomings that should be corrected before the publication of the work:
1) Key words: should not be repeated with the words used in the title.
2) The abstract is a separate part of the work (it can be published independently of the rest of the work). You shouldn't use abbreviations in it. Again - in the introduction all abbreviations should be re-explained.
3) Introduction: I don’t have any doubts about the scope of work and its novelty. However, Authors did not clearly set the aim of the work.
4) Results and discussion, line 255 - 256: The authors state that GSH improve photosynthesis. However, they did not post such results anywhere.
5) MM section: Are the Authors convinced that 3 replications are sufficient for such measurements? And are they convinced, that parametric test is suitable for these data?
6) As already mentioned above - the workflow should be improved.
I would like to underline that my remarks are auxiliary and not undertake the quality and importance of the paper.
Round 2
Reviewer 1 Report
The revised manuscript is improved substantially and satisfied most concerns. I recommend this article for publication in the Plants journal